# MLK4 as an immune marker and its correlation with immune infiltration in Cervical squamous cell carcinoma and endocervical adenocarcinoma(CESC)

**Meng Gong[1], Fujin Shen[1], Yang Li[1], Lei Ming[2], Li Hong[1]***

**1** Gynecology Department, Renmin Hospital of Wuhan University, Wuhan, China, **2** Reproductive Medicine Center, Renmin Hospital of Wuhan University, Wuhan, China

* drwhdxrmyyhongli@163.com

**Data Availability Statement:** All relevant data are within the manuscript and its Supporting Information files.

## Abstract

Mixed pedigree kinase 4 (MLK4) is a member of the serine/threonine kinases mixed pedigree kinase (MLKs) family. Few reports on immune-related targets in Cervical squamous cell carcinoma and endocervical adenocarcinoma (CESC), and the role of MLK4 in cervical cancer remains to be studied. The expression of MLK4 in CESC was analyzed by TCGA database containing 306 CESC tissues and 3 peritumoral tissue samples, and the effect of MLK4 on immune invasion was evaluated using the Deseq2 package(Benjamini-Hochberg corrected p-value < 0.05 and log2 fold change $\geq |2|$). Tissue microarray was used to verify the expression of MLK4 in CESC patients, and it was found that MLK4 was significantly overexpressed in CESC, and significantly correlated with WHO grade. Multiple analysis algorithms revealed that the high expression of MLK4 was negatively correlated with immune cell infiltration in CESC. Analysis showed that MLK4 expression was negatively correlated with the infiltration of various immune cells including CD8+T cells, and MLK4 mRNA expression was positively correlated with immune checkpoints PD-L1,CTLA4, LAG3, and negatively correlated with immune promotion genes CD86 and CD80. Furthermore, vitro assays were performed to investigate the biological characteristics of MLK4 in C33A cells. The EDU and transwell assays demonstrated that the decrease in MLK4 expression in C33A cells resulted in a decrease in cell proliferation and invasion. The silencing of MLK4 resulted in a significant increase in the expression of inflammatory cytokines IL-1β(p<0.05), TNF-α(p<0.01), and IL-6 (p<0.05). The results of cell assays indicate that knocking down MLK4 would inhibit the expression of established biochemical markers CEA, AFP and HCG. Hence, it is plausible that MLK4 could potentially exert a significant influence on the development and progression of Cervical cancer.

## 1. Introduction

Cervical squamous cell carcinoma and endocervical adenocarcinoma(CESC) has risen from the fourth most common cancer in the world to the third most common cancer in women,

**Funding:** The present study was financially supported by the Fundamental Research Funds for the Central Universities (grant number: 2042022kf1110).

**Competing interests:** The authors have declared that no competing interests exist.

according to 2020 statistics [1]. There are two main types of cervical cancer based on cell type: squamous cell carcinoma, identified in 80–90% of cases, and adenocarcinoma [2]. Squamous cell carcinoma arises from cells located in the ectocervix, whereas adenocarcinomas originate from the glandular cells found in the endocervix [2]. Different types of cervical cancer molecular biomarkers, including DNA methylation patterns, miRNA and specific proteins, are currently being studied to diagnose cervical cancer in patients [3]. In a recent report on protein markers, Zhe Wang et al. [4] used bioinformatics to screen ANXA2 as a diagnostic marker for cervical cancer and found that it was associated with the prognosis of cervical cancer patients [4]. Hui Ma et al. [5] found that MCM3 can be an important marker of cervical cancer [5].

Tumor microenvironment plays a crucial role in the occurrence and development of tumors. According to famous theories, there is a dynamic balance between tumor cells and immune cells in the development of diseases, and immune escape is almost the hallmark of all cancers [6]. During immune escape, tumor cells release cytokines to protect abnormal cells from cytotoxic immune cells, promote the invasion of immunosuppressive cells or regulate the activation of immune checkpoints [7–9]. Although relatively high programmed cell death protein-1 (PD-1)/programmed cell death protein-ligand1 expression has been demonstrated in cervical tumors, the expression of immunosuppressive molecules prior to invasion has not been adequately studied [10, 11]. Therefore, exploring new protein markers related to immune invasion is helpful for the immunotherapy of cervical cancer.

To investigate immunological microenvironmental effects in tumors, researchers developed a new algorithm that uses the distinct features of cancer samples' transcription profiles to calculate the contents of tumor cells and the various invading normal cells, which can provide insights into the role of these cells in tumor progression and response to therapy. It is named Estimation of STromal and Immune cells in MAlignant tumor tissues using Expression data (ESTIMATE) [12, 13]. ESTIMATE focuses on stromal and immune cells, which make up the majority of non-tumor components in tumor samples, and identifies specific signals associated with stromal and immune cell infiltration in tumor tissue [13]. The levels of infiltrating stromal and immune cells were predicted by computing stromal and immune scores by performing a single-sample gene set enrichment analysis (ssGSEA), which serves as the basis for estimating the ESTIMATE Score of tumor purity in tumor tissue [14–16]. In cervical cancer research, ESTIMATE has been used to analyze gene expression data from cervical cancer tissues and to investigate the association between the tumor microenvironment and patient outcomes [17], CXCR3 axis may be a new therapeutic target for cervical cancer [17]. Yali Deng et al. [18] also used ESTIMATE algorithm to calculate tumor purity of cervical cancer.

MLK4 is a member of the serine/threonine kinases that belongs to mixed pedigree kinase (MLKs) family [19]. The MLK4 gene (MAP3K21/KIAA1804) is frequently mutated and overexpressed in various types of human cancer [20]. In recent years, the role of MLK4 in cancer cell biology and tumor progression has been reported. Mehlich D et al. [19] reported that MLK4 regulates DNA damage and promotes chemical resistance in triple negative breast cancer [19]. Their reports [21] demonstrated that MLK4 promotes the migration and invasion in breast cancer cells [21]. Kim SH et al. [20] demonstrated that MLK4 regulates NF-κB activation in gliomas [20]. Despite growing knowledge about the role of MLK4 in tumorigenesis, the role of MLK4 in cervical cancer has not been reported.

In this study, we analyzed the expression profile data of patients with cervical cancer using ESTIMATE to analyze the differentially expressed genes between patients with high and low immune invasion, and co-screened them with genes with high expression in cervical cancer compared with those in the peritumoral to identify MLK4 as a possible molecular marker of cervical cancer immune invasion. The function of MLK4 in cervical cancer was confirmed by immunohistochemistry and in-depth bioinformatics analysis.

## 2. Materials and methods

### 2.1 Gene expression analysis of cervical cancer RNA-seq data from TCGA

RNA sequencing (RNA-Seq) data from Cervical squamous cell carcinoma and endocervical adenocarcinoma (CESC) patients were obtained from the TCGA data portal (https://cancergenome.nih.gov/.), which contained 306 CESC tissues and 3 peritumoral tissue samples up to September 18, 2021. The Deseq2 package was used to identify differentially expressed genes (DEGs) between tumor and normal samples (Benjamini-Hochberg corrected p-value 0.05 and log2 fold change |2|).

### 2.2 Functional analysis

Functional annotation analyses: GO (Gene Ontology) and KEGG (Kyoto Encyclopedia of Genes and Genomes) were performed using the Database for Annotation, Visualization, and Integrated Discovery (DAVID; version 6.8; https://david.ncifcrf.gov/), using Homo sapiens genes as background. Terms with Benjamini-Hochberg corrected p-values < 0.05 were determined as enriched.

### 2.3 Survival analysis in Cervical cancer

Gene Expression Profiling Interactive Analysis (GEPIA) server was used to evaluate whether increased expression of the selected DEGs was associated with a poorer prognosis in cervical cancer. Samples with increased expression of selected DEGs were compared with their low expression samples. Kaplan-Meier survival curves were built and then compared using a log-rank test with multivariate analysis. Differences resulting in a p-value < 0.05 were considered significant.

### 2.4 Grouping and screening of immune infiltration

The infiltration index of each patient was analyzed using CESC data from TCGA database. We used the R package IOBR to select the ESTIMATE method to calculate the immune infiltration score for each sample. The top 50 patients with the highest and lowest immune infiltration scores were grouped for differential gene analysis, and the previous differential genes were co-screened.

### 2.5 Analyze the effect of MLK4 on immune cell infiltration in Cervical cancer

We downloaded the unified CESC data set from the UCSC database(http://genome.ucsc.edu/), and further extracted the expression data of MLK4 gene in each sample. The source of the selected samples was as follows: Log2 (x+0.001) transformation was performed for each expression value of Primary Tumor and Primary solid Tumor, and eight immune scoring methods were used: ESTIMATE [22] quanTIseq [23], TIMER [24], IPS [25], MCPCounter [13], xCell [26], EPIC [27] and CIBERSORT [28] calculated the effect of MLK4 on immune infiltration in cervical cancer.

### 2.6 Immune checkpoint correlation analysis

The correlation analysis between immune regulatory genes and the immune checkpoint was conducted using the Sangerbox tool (version 3.0; http://vip.sangerbox.com/home.html). In this study, the CESC data from The Cancer Genome Atlas (TCGA) was utilized to conduct a Log2 (x+0.001) transformation. The statistical analysis employed the Pearson correlation coefficient.

## 2.7 Immunohistochemistry

A cervical cancer pathology microarray was obtained from Spector Dotarry lab Biotechnology. The expression of MLK4 in tumor tissues was detected by immunohistochemical method. Paraffin-embedded tumor tissue sections with a thickness of 4 μm were obtained. The tissue slice underwent a deparaffinization process by being placed at a temperature of 70˚C for 90 minutes. The endogenous peroxidase activity is then blocked at room temperature by a 5–10 minute incubation in 3% H2O2. The sample should be incubated with a 5% bovine serum albumin (BSA) solution for a duration of 20 minutes at room temperature. The tissue slice were subsequently incubated overnight at a temperature of 4˚C with an anti-MLK4 antibody (GXP326155, 1:200 dilution, GENXSPAN, USA). Following two washes with a tris buffered saline (TBS) solution containing 0.025% Triton X-100, the tissue slice was exposed incubated with an HRP-conjugated secondary antibody II for 1 hour. Subsequently, the sample was stained with DAB (Biosharp, BL732A) at room temperature for approximately 10 minutes. Tissue section images were visualized with a Nikon microscope equipped with NIS component software and quantified with 3DHISTECH Quant Center 2.2 software. The H-Score can be calculated by evaluating the proportion of positive and negative pixels. The positive area ratio is defined as the ratio of the positive area to the total tissue area. Positive area ratio = Positive area/total tissue area. The average was taken as the positive expression rate of the patient, and the positive expression rate of MLK4 >50% was defined as positive. All studies were approved by the Ethics Committee of Wuhan University People's Hospital (NO.20231X089).

## 2.8 cell culture

C33A cell line was purchased from Guangzhou Cellcook Biotech (Guangzhou, China). All cells were cultured in Dulbecco's Modified Eagle medium (DMEM, HyClone) with 10% fetal bovine serum (Gibco, USA) at 37˚C with 95% air humidity and 5% CO2.

## 2.9 Small interfering RNAs and cell transfection

MLK4 small interfering RNA (siRNA) was synthesized: siRNA-01, GGAAAGAUGCUCAGAGAG AUU; MLK4-siRNA-02, AGGAGAAGCCCAAGGAAUU, MLK4-siRNA-03, AGAAGAAACGAGAG GGAAUU. The transfection of negative control (NC) and small interfering RNA (siRNA) was performed using Lipofectamine RNAiMAX. The complete list of siRNA and primers can be found in *S1 Table*.

## 2.10 Western blotting

The cellular protein was extracted by the addition of 2× Laemmli sample buffer and a protease inhibitor. The BCA protein assay kit was used to calculate the concentration of protein. Lysates were heated to 95 ˚C for 5 min. A quantity of 20–30 μg of proteins was subjected to separation using SDS-PAGE gel and subsequently transferred onto 4.5 μm PVDF membranes. Subsequently, the sample was subjected to a blocking step using 5%milk for 1 hour. Then, the sample was incubated overnight at a temperature of 4˚C with an anti-MLK4 antibody. MLK4 antibody was purchased from GENXSPAN (GXP326155), IL1β antibody was purchased from Santacruz (sc-12742),TNF-α antibody was purchased from Abcam(ab183218).

## 2.11 Quantitative reverse-transcription PCR (qRT-PCR)

The extraction of total RNA from cultured cells was performed using TRIzol reagent according with the established protocol. Subsequently, the 700ng RNA was reversed transcription using the GoScriptTM reverse transcription Mix. Quantitative real-time polymerase chain reaction

(qRT-PCR) was conducted utilizing the SYBR Green Master Mix. Beta-actin primer a was used as an internal control. The primer sequences of this study can be found in *S1 Table*.

## 2.12 EDU assay and Transwell assay

The EdU proliferation assay was conducted subsequent to the transfection of plasmids in C33A cells. According to the instructions provided by the manufacturer of the Abbkine EdU kit. The cells were subjected to incubation with an EdU agent at a concentration of 100 μM for a duration of 2 hours. Dapi was diluted in phosphate-buffered saline (PBS) at a ratio of 1:500. The images were captured using a Lecia SP8 confocal microscope. Cell proliferation rate was calculated by dividing the number of active dividing cells (red) by the total number of cells (blue).

Transwell assay was utilized to evaluate cell invasion by using sterile Transwells® inserted with a polycarbonate membrane of 8.0 μm aperture. Following a 48-hour period of transfection, a total of $5 \times 10^4$ cells were introduced into the upper chamber without the presence of fetal bovine serum (FBS). The cells were underwent incubation at a temperature of 37˚C for 24h. Following this, the cells were fixed and subsequently stained using hematoxylin. Cells in five random fields were counted under a microscope.

## 3. Result

### 3.1 Differential analysis of gene expression in cervical cancer

Firstly, the cervical cancer data in TCGA database were divided into cancer group (T) and normal group (N). To determine the gene differences between the two groups of cervical cancer, the Limma package was used to analyze the differential expression of genes in cervical cancer patients, and A total of 2970 differential gene expressions were obtained *(Fig 1A, S2 Table)*, among which 955 genes were highly expressed, 2015 for lower expression (|FC|$\geq$2, p<0.05), indicating that there were gene expression differences between the cancer group and the normal group of cervical cancer patients. To investigate the molecular markers associated with immune infiltration in cervical cancer patients, we analyzed the RNAseq data of cervical cancer with ESTIMATE to calculate the immune infiltration score of cervical cancer patients *(Table 1, S3 Table)*. Patients with Top50 High immune infiltration score (high) and patients with a low immune infiltration score (low) were screened. PCA analysis showed that the genes of the two groups were completely separated *(Fig 1B)*, in which PCA1 was 10.18% and PCA2 was 4.80%. Using Limma gene expression analysis for 5614 different genes expression *(Fig 1C, S4 Table)*, of which 2589 high gene expression, 3025 genes lower expression (|FC|$\geq$1, p<0.05), indicating that there were significant differences in gene expression among different groups of cervical cancer patients with different immune infiltration.

### 3.2 Screening and analysis of immune-related differential genes in cervical cancer

In order to obtain molecular markers related to cervical cancer immunity and prognosis, we co-screened the highly expressed genes in T vs N group and the highly expressed genes in low-I (low-immunescore) vs high-I group (high-immunescore) *(Fig 2A)*, and carried out GO and KEGG analysis on these genes. GO analysis showed that the gene set was mainly related to the anatomical structure maturation *(Fig 2B)*, whereas KEGG analysis demonstrated that these genes were primarily associated with Natural Killer cell-mediated cytotoxicity *(Fig 2C)*. Gene-MANIA (http://genemania.org) was further used to analyze gene interactions *(Fig 2D)* for subsequent marker screening.

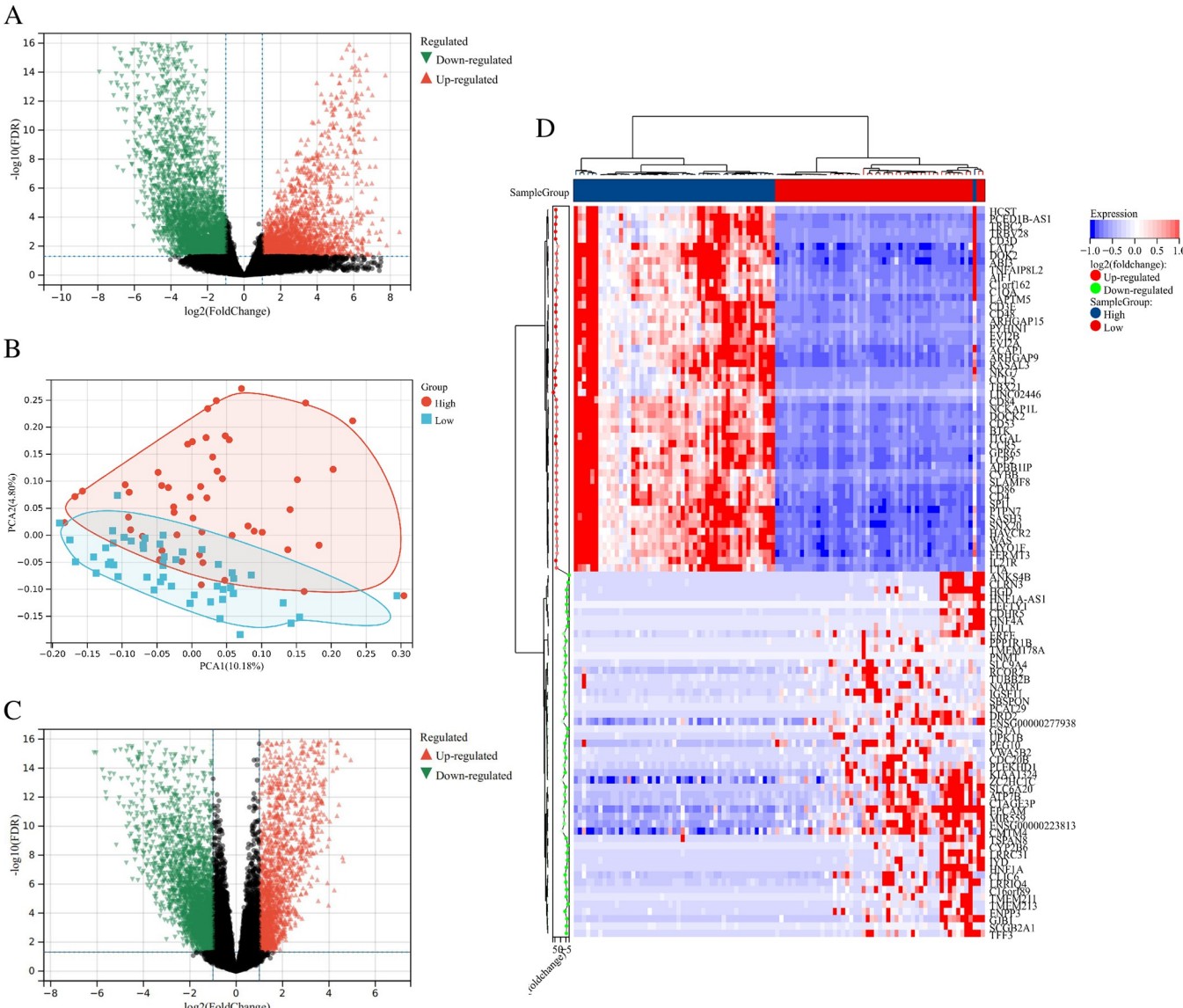

**Fig 1. Differential analysis of gene expression in cervical cancer.** A. Differential gene volcano map of cervical cancer; B. PCA of cervical cancer patients of various immune infiltration; C. volcanic map of cervical cancer patients with differing immune infiltration; and D. differential gene heat map of cervical cancer patients with different immune infiltration.

### 3.3 Screening and validation of immune-related molecular markers in cervical cancer

Among the candidate genes, we excluded candidates of small RNA, mainly focused on genes encoding proteins, and analyzed the disease-free survival and overall survival of cervical cancer patients with Kaplan–Meier curves *(Fig 3A)*. The analysis revealed that MLK4 gene expression decreased the disease-free survival of patients with cervical cancer *(Fig 3A,* DFS, Logrank p = 0.014, p(HR) = 0.016*)*. AGEPIA analysis of MLK4 expression in patients with cervical cancer revealed that MLK4 was significantly overexpressed in these patients *(Fig 3B)*.

Using a tissue microarray, the expression of MLK4 in cervical cancer and adjacent tissues was confirmed. According to the results of HE *(Fig 3C)* and IHC, the proportion of cervical

**Table 1. Patients samples with the top10 immune scores in the ESTIMATE analysis.**

| ID | Stromal Score | Immune Score |
|---|---|---|
| TCGA-DG-A2KM-01 | 494.1996408 | 3068.176131 |
| TCGA-MY-A5BE-01 | -64.96949764 | 2832.891361 |
| TCGA-R2-A69V-01 | 154.8617328 | 2657.365762 |
| TCGA-MA-AA42-01 | -570.2167874 | 2456.375333 |
| TCGA-EA-3QD-01 | 161.7264332 | 2120.509177 |
| TCGA-IR-A3LH-01 | 543.2190549 | 2098.288123 |
| TCGA-C5-A7CG-01 | -531.2331856 | 2071.684722 |
| TCGA-VS-A9UT-01 | 119.0178517 | 1926.750131 |
| TCGA-VS-A9UD-01 | -1121.828106 | 1878.365157 |
| TCGA-VS-A9UH-01 | 234.6050061 | 1828.731203 |

cancer tissues with high MLK4 expression was significantly greater than that of adjacent tissues (*Fig 3D)*. The expression of MLK4 was significantly correlated with the WHO Grade of patients *(*p<0.05, *Table 2)*. In conclusion, we found that MLK4 may serve as a molecular marker related to cervical cancer immunity.

## 3.4 MLK4 mRNA expression level is related to immune cell infiltration

To further investigate the effect of MLK4 on cervical cancer immunity, we used a variety of immune scoring algorithms to examine the effect of MLK4 on the infiltration and proportion of immune cells in cervical cancer. According to ESTIMA results, MLK4 expression was significantly negatively correlated with immune cell infiltration score *(Fig 4A*, p<0.0001*)*. Using IPS analysis, it was found that MLK4 expression was significantly negatively correlated with MHC (major histocompatibility complex) and EC(Effect cell) expression *(Fig 4B*, p<0.0001*)*. After analyzing the effect of MLK4 on immune cell infiltration using a variety of immune cell infiltration algorithms *(Fig 4C)*, we found that MLK4 expression was negatively correlated with CD8 +T cell expression. CIBERSORT and xCELL were used to verify the results, and identical results were obtained *(Fig 4D*, p<0.0001*)*, indicating that in cervical cancer, MLK4 expression affects the infiltration of a variety of immune cells and the number of CD8+T cells significantly.

## 3.5 MLK4 mRNA expression is related to immune checkpoint gene expression

Overexpression of immune checkpoint is one of the causes of tumor immune escape. Therefore, we explored the correlation between MLK4 expression and the expression of immune checkpoint genes and immune regulatory genes in cervical cancer. We verified the correlation between MLK4 gene and cervical cancer immune checkpoint expression. The results demonstrated that MLK4 was associated with immune checkpoints CD274, CTLA4, LAG3, and IDO1. These immune checkpoints were positively correlated with immune-promoting genes, such as CD86 and CD80 *(Fig 5A and 5B*, p<0.05), and significantly negatively regulated, indicating that MLK4 may affect the invasion of immune cells in cervical cancer by regulating the expression of immune checkpoints and immune-related genes, leading to immune escape of tumors and affecting the survival of patients. To verify these bioinformatics results, qPCR was conducted on C33A cells. qPCR results demonstrated that inhibition of MLK4 led to significant down-regulation of the immune checkpoint gene CD274(p<0.05)CTLA4 (p<0.01)and IDO1(p<0.01), and up-regulation of the immune-promoting genes CD80 (p<0.001) and CD86(p<0.01) in C33A cells in against the control *(Fig 5C)*.

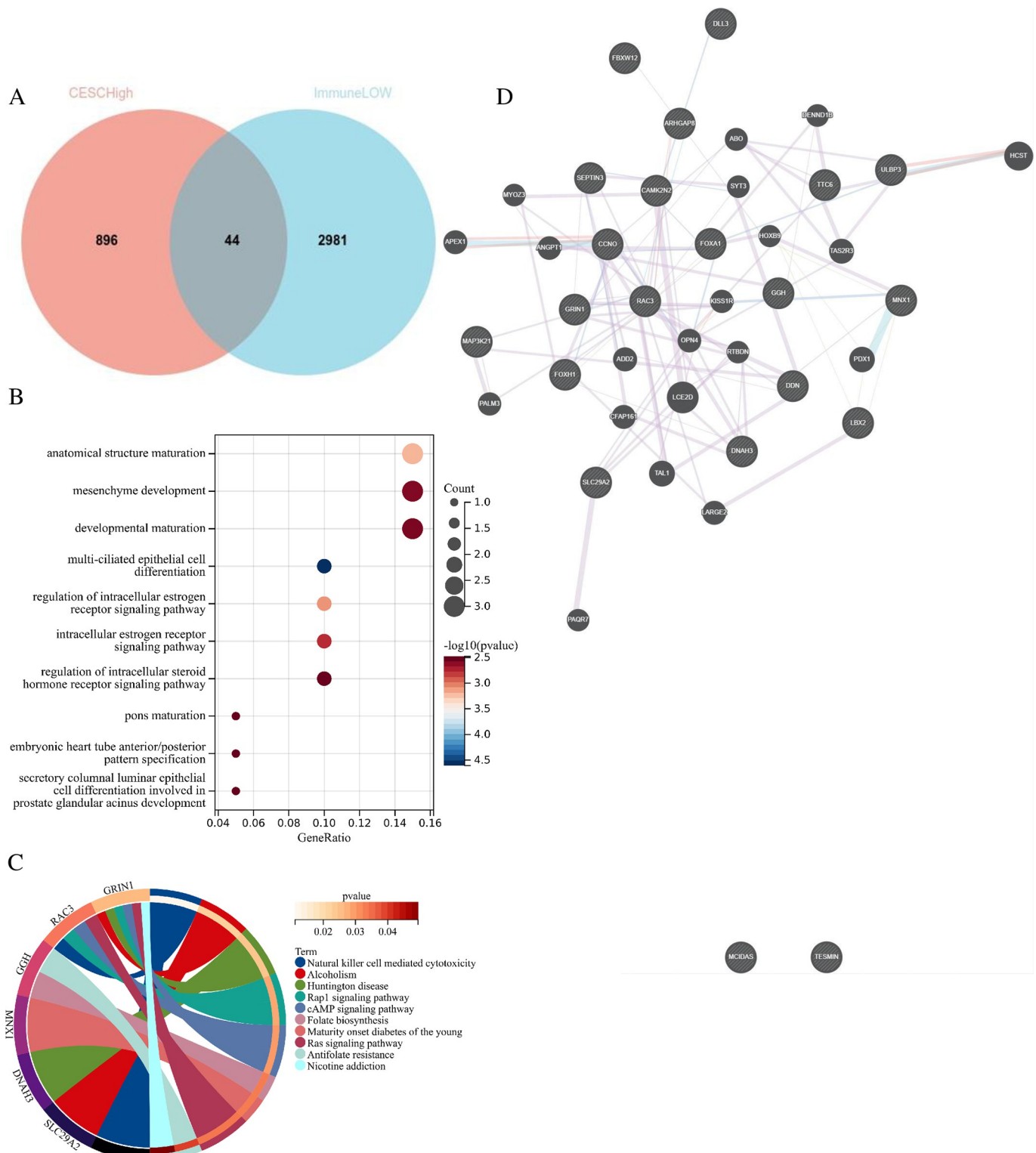

**Fig 2. Screening and analysis of immune-related differential genes in cervical cancer.** A. Venn diagram of highly expressed genes in cervical cancer and low immune infiltration groups; B.GO analysis of intersection genes; C.KEGG analysis of intersection genes; D. Gene interaction analysis of intersection genes.

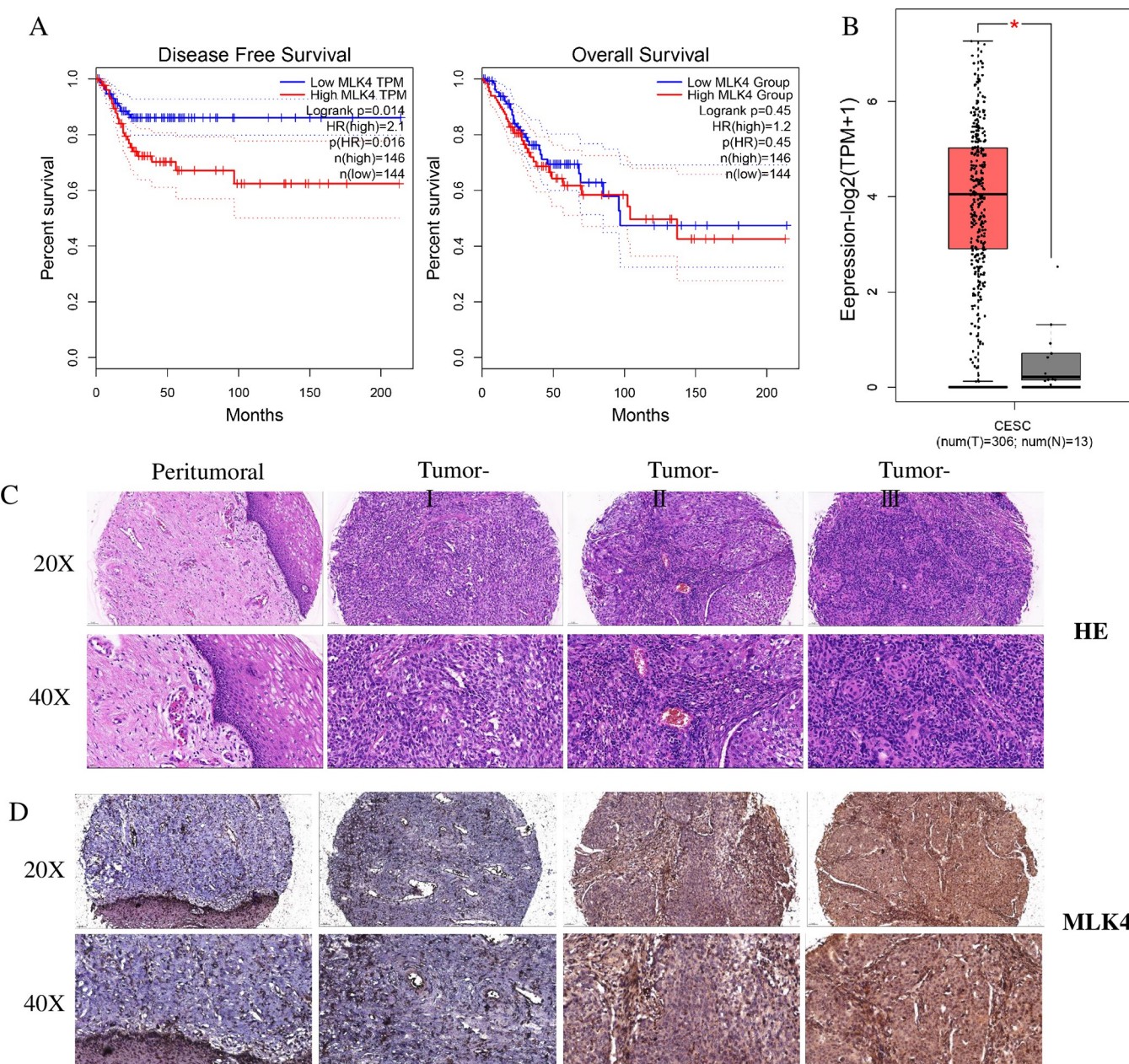

**Fig 3. Screening and validation of immune-related molecular markers in cervical cancer.** A. Kaplan–Meier curves showing disease free survival (DFS) and overall survival (OS) recorded in all CESC patients of low MLK4(blue line) and high MLK4 (red line), recorded in TCGA array data. B. Expression of MLK4 in cervical cancer and Peritumoral; C. HE staining of cervical cancer tissue microarray; D. IHC staining of MLK4 in cervical cancer tissue microarray.

## 3.6 Inhibition of MLK4 significantly promoted the proliferation and invasion in Cervical cancer

To identify the biological behavior of MLK4 in cervical cancer, we conducted in vitro assays of C33A cells. The effectiveness of MLK4 siRNA was confirmed by quantitative real-time polymerase chain reaction (qRT-PCR) and western blotting; si-MLK4-02 significantly suppressed MLK4 expression in C33A cells *(Fig 6A and 6B,* p<0.01*)*. Immunofluorescence on C33A cells revealed that MLK4 is mainly found in the nucleus *(Fig 6C)*.CCK-8 and EDU proliferation

**Table 2. Correlation analysis between MLK4 and clinicopathological information.**

| Features | No. of patients | MLK4 expression | | p value |
|---|---|---|---|---|
| | | low | high | |
| All patients | 80 | 36 | 44 | |
| Age (years) | | | | 0.1772 |
| <61 | 36 | 15 | 26 | |
| ≥61 | 44 | 21 | 18 | |
| Grade | | | | 0.0328* |
| I | 3 | 3 | 0 | |
| II | 55 | 27 | 28 | |
| III | 22 | 6 | 16 | |
| (N) | | | | 0.0986 |
| N0 | 63 | 25 | 38 | |
| N1 | 17 | 11 | 6 | |
| T | | | | 0.3287 |
| T1 | 69 | 33 | 36 | |
| T2 | 11 | 3 | 8 | |
| Tumor size | | | | 0.2481 |
| <5cm | 77 | 36 | 41 | |
| ≥5cm | 3 | 0 | 3 | |
| Stage | | | | 0.7335 |
| IA | 1 | 0 | 1 | |
| IB | 68 | 32 | 36 | |
| IIA | 8 | 3 | 5 | |
| IIB | 3 | 1 | 2 | |

assays revealed that inhibition of MLK4 promoted C33A cell proliferation significantly compared to the control (*Fig 6D and 6E*, p<0.05,). Transwell assays were conducted to detect the invasive ability of MLK4 in cervical cancer(*Fig 6D and 6E*,p<0.05,).Cell invasion in C33A cells was decreased as a result of MLK4 downregulation(*Fig 6F*, p<0.001,). These findings confirm that MLK4 is closely involved in cell proliferation and invasion in Cervical cancer. As revealed by the EDU and transwell assays, the downregulation of MLK4 in C33A cells led to a reduction in cell proliferation and invasion.

### 3.7 Silencing of MLK4 reduces the expression of inflammatory cytokines and classical biochemical markers in Cervical cancer

In the context of immunosuppression and autoactivation signals, tumor-secreted cytokines and growth factors are capable of promoting cell growth, survival, and invasion [22]. The interaction and effect of MLK4 with immunological molecules(TLR4, TNF-α, IFN-γ, IL-6, IFN-γ and TGF-β) were summarized in *Table 3* [29–32]. In addition, we examined the expression of inflammatory cytokines, which are associated genes with immune cell infiltration IL-1β, TNF-α and IL-6. As shown in *Fig 7A–7C*, silencing MLK4 in C33A cells significantly increased the expression of IL-1β(p<0.05), TNF-α(p<0.01)and IL-6 (p<0.05)compared to the control. Western blot analysis confirmed that knocking down MLK4 could increase the expression of IL- and TNF (*Fig 7D*). It suggests that MLK4 molecules may modulate the tumor microenvironment as tumor-derived molecules.

Emerging evidence indicates that carcinoembryonic antigen (CEA), alpha-fetoprotein (AFP), and human chorionic gonadotropin (HCG) are established biochemical markers in

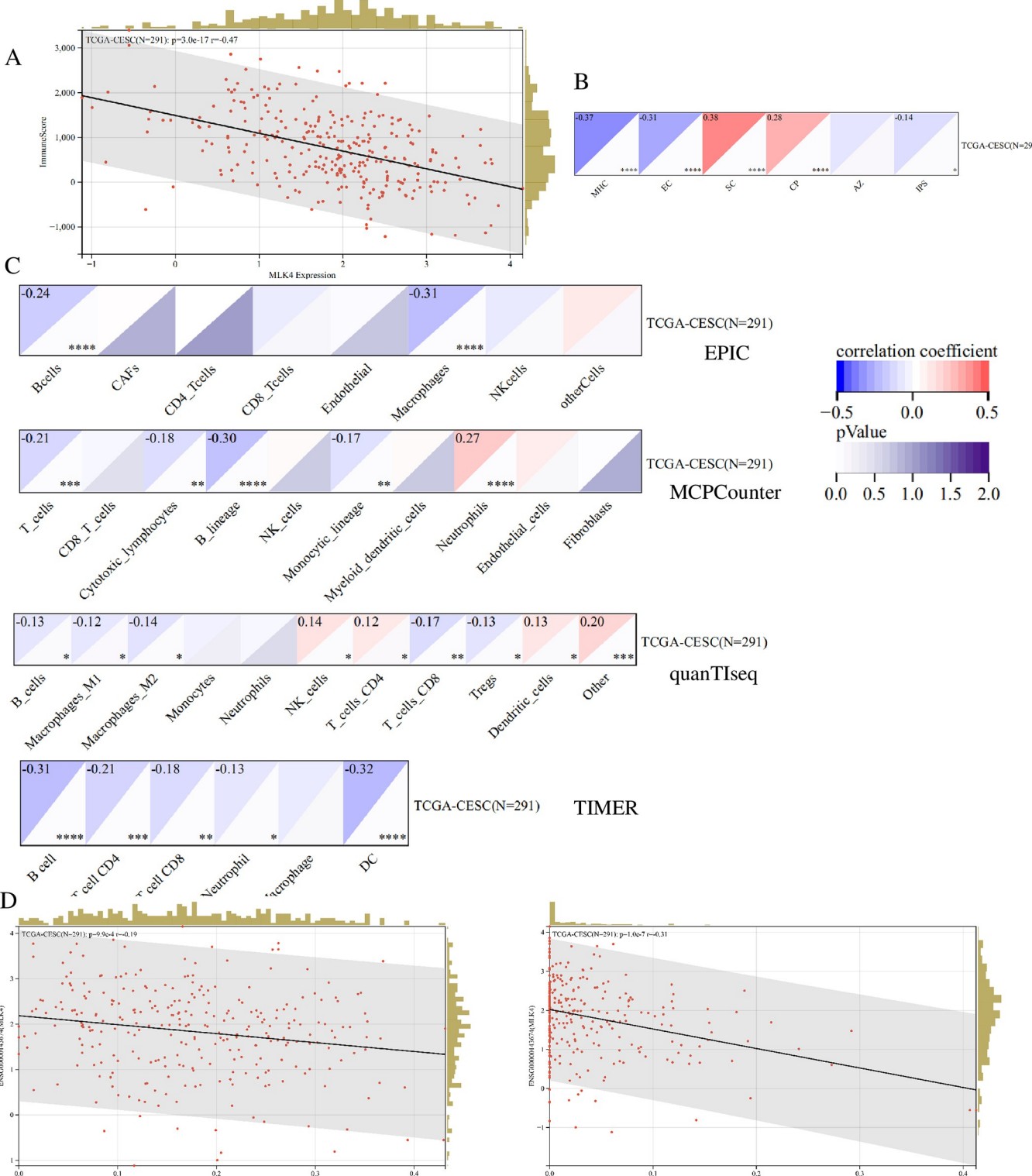

**Fig 4. MLK4 mRNA expression level is related to immune cell infiltration.** A. ESTIMATE was used to analyze the correlation between MLK4 and immune cell infiltration in cervical cancer B. The correlation between MLK4 and immune program was analyzed by IPS; C. Four different algorithms were used to analyze the correlation between MLK4 and immune cell infiltration in cervical cancer; D. Two algorithms were used to analyze the correlation between MLK4 and $CD8^{+}T$ cell infiltration in cervical cancer(*P < 0.05, **P < 0.01, ***P < 0.001,****P < 0.0001).

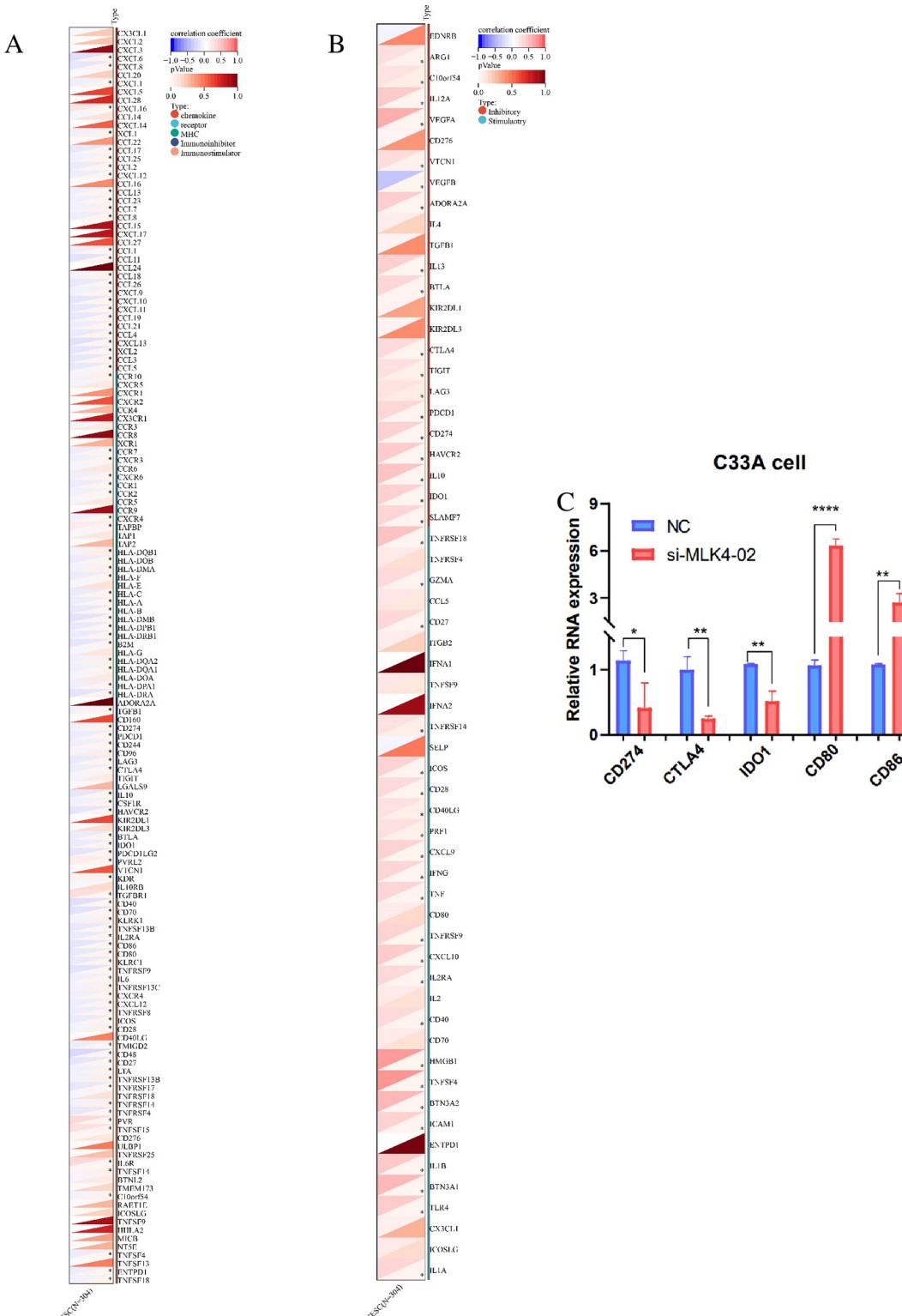

**Fig 5. MLK4 mRNA expression is related to immune checkpoint gene expression.** A. Correlation analysis between MLK4 and immune-related factors in cervical cancer; B. Correlation analysis between MLK4 and immune checkpoint in cervical cancer C. qPCR assays demonstrated that inhibition of MLK4 led to regulation of immune checkpoint CD274CTLA4IDO1 and immune-promoting genes CD80 and CD86 in 33A cells relative to the control(means ± SD; *P < 0.05, **P < 0.01, ***P < 0.001).

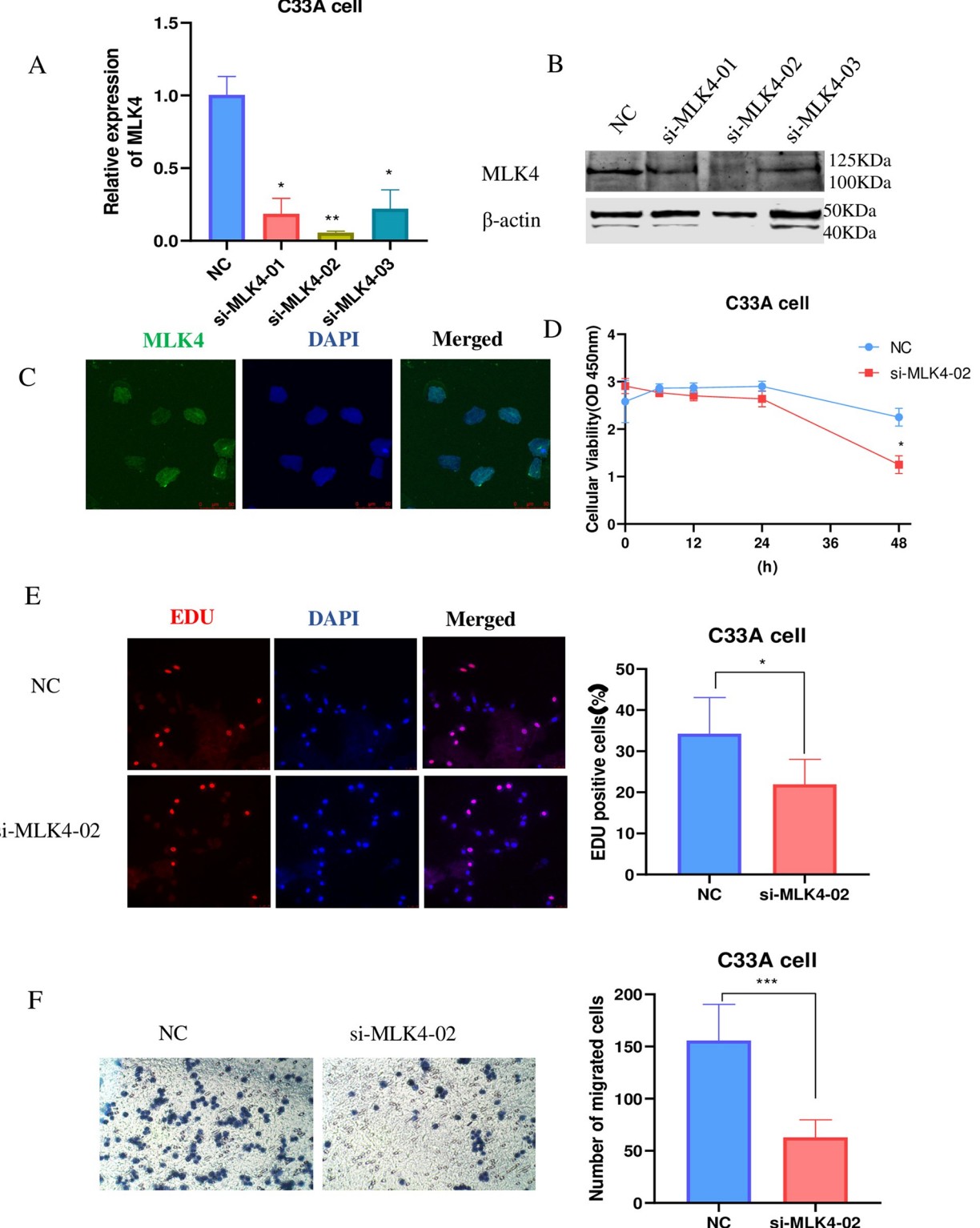

**Fig 6. Inhibition of MLK4 significantly promoted the proliferation and invasion in Cervical cancer.** A and B. Three siRNAs were designed to target MLK4. The effect of siRNAs or plasmids were detected by qRT-PCR and Western blot. C. Immunofluorescence showed that MLK4 mainly located in nucleus. D and E. CCCK-8 and EdU assays were performed to assess the proliferative potential of MLK4 in C33A cells. F. Transwell assays were conducted to determine the invading potential of MLK4 in C33A cells.(The data represent means ± SD; *P < 0.05, **P < 0.01, ***P < 0.001, 25uM, Representative images are shown).

**Table 3. The interaction and effect of MLK4 with immunological molecules.**

| Immunological Molecules | Interaction with MLK4 | Effect | PMID |
|---|---|---|---|
| Toll-like receptor 4 (TLR4) | Directly activates MLK4 | Induces pro-inflammatory cytokine production, activates NF-κB, and promotes dendritic cell maturation and antigen presentation | [29] |
| Tumor necrosis factor alpha (TNF-α) | Activates MLK4 via TNFR1 | Induces apoptosis and inflammation, promotes macrophage activation and cytokine production, and enhances T cell activation and proliferation | [29] |
| Interleukin-1 type 1 receptor(IL-1R1) | Activates MLK4 via IL-1R1 | Induces pro-inflammatory cytokine production, activates NF-κB and MAPK pathways, and promotes neutrophil recruitment and activation | [30] |
| Interferon gamma (IFN-γ) | Activates MLK4 | Induces pro-inflammatory cytokine production, enhances antigen presentation and T cell activation, and promotes macrophage polarization towards M1 phenotype | [31] |
| Interleukin-6 (IL-6) | Indirectly activates MLK4 via STAT3 | Induces pro-inflammatory cytokine production, promotes B cell differentiation and antibody production, and enhances Th17 cell differentiation and autoimmunity | [31] |
| Interleukin-17receptor B (IL-17RB) | MLK4 specifically phosphorylates IL-17RB at Y447 | Induces pro-inflammatory cytokine production, IL-17RB forms a homodimer and recruits MLK4, a dual kinase, to phosphorylate it at tyrosine-447 upon treatment with IL-17B | [32] |

plasma and tumors of patients with gynecological malignant tumors, according to emerging evidence [33]. Specifically, patients with ovarian and endocervical mucinous adenocarcinoma had the highest levels of CEA [34]. Alpha-fetoprotein was frequently elevated in patients with ovarian germinoma, stromal tumors, and large-cell nonkeratinizing cervical cancer. Patients with severe cystadenocarcinoma of the ovary and keratosis squamous cell carcinoma of the cervix had the highest HCG concentrations [34]. C33A cells transfected with NC and si-MLK4 were analyzed by RT-PCR to assess the biological relationship between MLK4 and conventional biochemical markers. Fig 7E demonstrates that MLK4 knockdown inhibited the expression of CEA(p<0.05), AFP(p<0.05) and HCG(p<0.05, *Fig 7E*). MLK4 could play an important role in cervical cancer.

## 4. Discussion

Here, we report a study on screening and validation of immune-related biomarkers for cervical cancer. The expression of MLK4 mRNA was significantly increased in cervical cancer compared with adjacent normal tissues. MLK4 mRNA level is associated with immune infiltration status, disease-free survival and Grade stage in patients with cervical cancer. To our knowledge, this is the first study to report an association between increased MLK4 mRNA levels and immune infiltration in cervical cancer patients.

An important aspect of this study is that MLK4 is associated with different levels of immune infiltration. By downloading the data of cervical cancer RNA-Seq from TCGA database and using the ESTIMATE calculation tool [13], we first observed the set of genes negatively correlated with immune cell infiltration in patients with different groups of immune cell infiltration, and according to the survival analysis and protein expression in cervical cancer patient samples, MLK4 may be used as a molecular marker related to cervical cancer immune infiltration. In cervical cancer tissue microarray, MLK4 expression was significantly higher in cancer tissues than in adjacent tumor tissues, and was correlated with the Grade stage of patients. According to the analysis results, we used a variety of immune cell infiltration algorithms to explore the correlation between MLK4 mRNA expression and immune infiltration in cervical cancer. The analysis results showed that MLK4 was significantly negatively correlated with a variety of immune cells in cervical cancer, including CD8+T cells. The correlation between MLK4 mRNA expression and immune checkpoint and immune-related factors was further explored. It was found that MLK4 mRNA expression was significantly positively correlated with immune checkpoint PD-L1, CTLA4, LAG3etc, which were related to the regulation of T cells in cervical cancer. Genes related to immune promotion, such as CD86 and CD80, have a

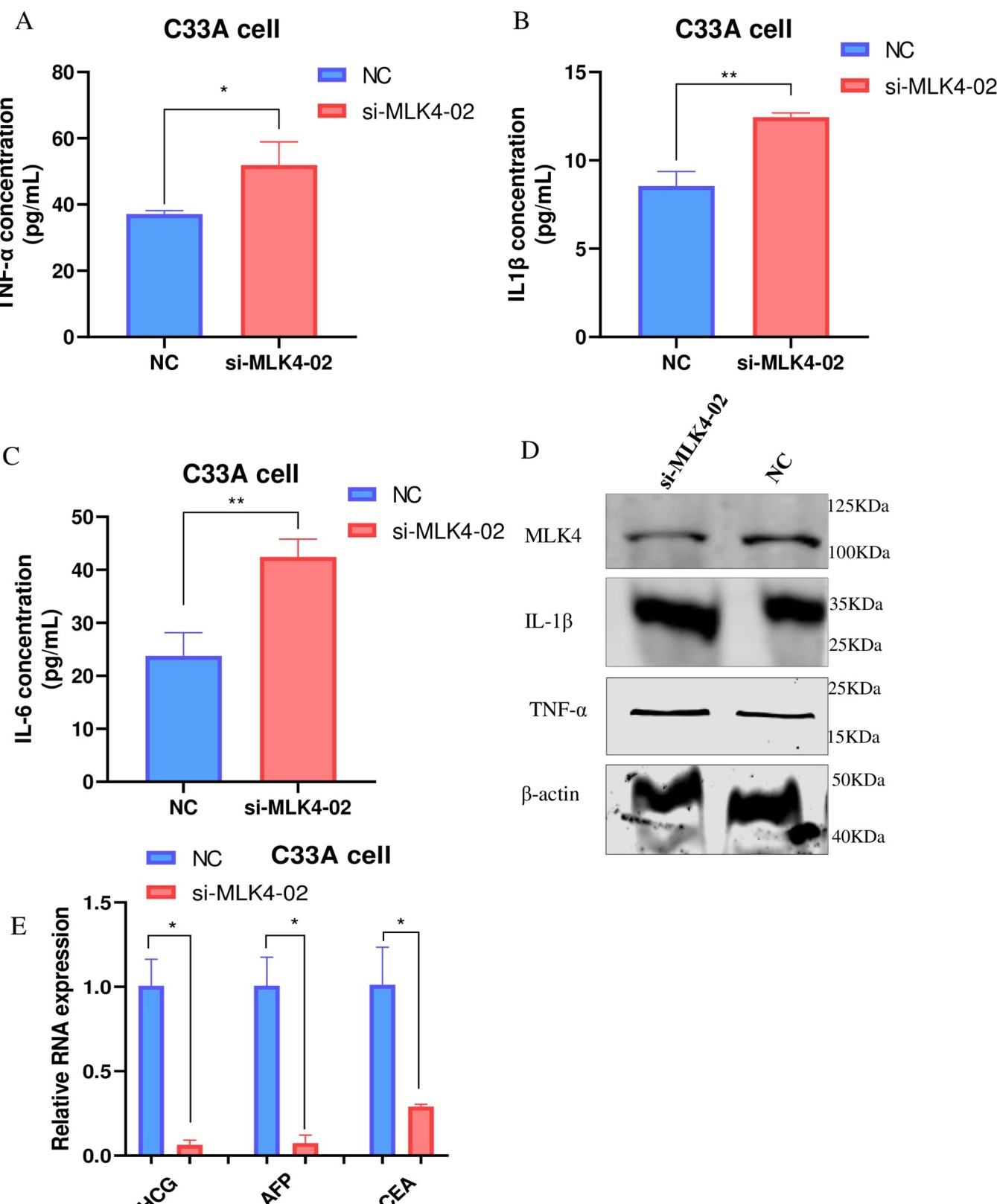

**Fig 7. Silencing of MLK4 reduces the expression of inflammatory cytokines and classical biochemical markers in Cervical cancer.** A-C. Elisa assays were conducted to analyze the expression of the inflammatory cytokines TNF-α, IL-βand IL-6 in C33A cells expressing MLK4. The levels of the inflammatory

cytokines TNF- and IL- were determined by western blotting. E. qPCR assays demonstrated that inhibition of MLK4 was associated with classic biochemical markers HCG, AFP and CEA (means ± SD; *P < 0.05, **P < 0.01, ***P < 0.001).

significant negative regulatory relationship, indicating that MLK4 may affect the infiltration of immune cells in Cervical cancer by regulating the expression of immune checkpoints. In this study, we conducted in vitro assays to explore the biological behavior of MLK4 in C33A cells. As revealed by the EDU and transwell assays, the downregulation of MLK4 decreased cell proliferation and invasion in C33A cells. The qPCR results indicate that knocking down MLK4 would inhibit the expression of the established biochemical markers CEAAFP and HCG. MLk4 may play an important role in cervical cancer.

Recently approved a block of cervical cancer cytotoxic T lymphocyte associated antigen 4 (CTLA4) and programmed cell death protein 1 (PD1) of a variety of therapeutic antibodies, and other immune targeted clinical checkpoint blocker [22], but questions remain about the blocker's treatment effect and patient prognosis. The successful response to immunotherapy depends on the immune composition or "immune environment" of the tumor microenvironment (TME) [23, 24]. In addition, the presence of an increase in certain cell types has also been shown to be associated with increased survival in patients with various forms of cancer [13, 24]. Therefore, in this study, in order to explore the effectiveness of the use of cervical cancer therapeutic antibodies, after analyzing the influence of MLK4 on immune cell infiltration, the expression of MLK4, CTLA4, PD-L1 and other immune checkpoints was further explored, so as to provide a new molecular marker for cervical cancer immunotherapy.

## Supporting information

**S1 Table. Related siRNA and primers of this study.**
(XLSX)

**S2 Table. Differential expression of CESC.**
(XLSX)

**S3 Table. ESTIMATE_CESC_immunescore.**
(XLSX)

**S4 Table. Differential gene expression according to immunological group.**
(XLSX)

## Acknowledgments

The authors acknowledge the participants who generously gave their help on this study.

## Author Contributions

**Conceptualization:** Li Hong.

**Data curation:** Meng Gong, Fujin Shen.

**Formal analysis:** Meng Gong, Fujin Shen.

**Investigation:** Meng Gong.

**Methodology:** Meng Gong, Yang Li.

**Software:** Meng Gong, Yang Li.

**Supervision:** Lei Ming, Li Hong.

**Validation:** Lei Ming, Li Hong.

**Visualization:** Lei Ming.

**Writing – original draft:** Meng Gong.

**Writing – review & editing:** Lei Ming, Li Hong.

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
