## [Decision Letter · Decision Letter 0]

13 Jun 2023

PONE-D-23-02890MLK4 as an immune marker and its correlation with immune infiltration in Cervical squamous cell carcinoma and endocervical adenocarcinoma(CESC)PLOS ONE

Dear Dr. Li,

Thank you for submitting your manuscript to PLOS ONE. After careful consideration, we feel that it has merit but does not fully meet PLOS ONE’s publication criteria as it currently stands. Therefore, we invite you to submit a revised version of the manuscript that addresses the points raised during the review process.

We look forward to receiving your revised manuscript.

Kind regards,

Parikshaa Gupta

Academic Editor

PLOS ONE

- https://doi.org/10.18632/aging.102514

- https://doi.org/10.1111/ejh.13895

In your revision ensure you cite all your sources (including your own works), and quote or rephrase any duplicated text outside the methods section. Further consideration is dependent on these concerns being addressed.

4. Please keep your tables as part of your main manuscript and remove the individual files. Please note that supplementary tables (should remain/ be uploaded) as separate "supporting information" files

Reviewers' comments:

Reviewer's Responses to Questions

**Comments to the Author**

1. Is the manuscript technically sound, and do the data support the conclusions?

Reviewer #1: Partly

Reviewer #2: Partly

2. Has the statistical analysis been performed appropriately and rigorously? 

Reviewer #1: I Don't Know

Reviewer #2: I Don't Know

3. Have the authors made all data underlying the findings in their manuscript fully available?

Reviewer #1: Yes

Reviewer #2: Yes

4. Is the manuscript presented in an intelligible fashion and written in standard English?

Reviewer #1: No

Reviewer #2: No

5. Review Comments to the Author

Reviewer #1: 1.I recommend professional English editing

2.Abstract: Please include numbers such as sample sizes of TCGA and array data, p values etc.

3.Material and Method:

-Can you describe ESTIMATE briefly?

-How was IHC scored? What is a low or high IHC expression (table 2)? Please define them in the text.

-I cannot download supplementary data. Can you submit an IHC picture?

-Can you submit Kaplan-Meier curves?

-Can you make a table/diagram to summarize the interaction of MLK4 with immunological molecules?

-What is para tumor tissue?

-Can you give the exact number of SCC and adenocarcinoma samples?

-How many normal samples were used in TCGA and TMA data sets?

-What is small RNA, do you mean miRNA?

Reviewer #2: The authors have studied the utility of MLK4 as an immune marker and correlated the same with immune cell infiltration in cervical squamous cell carcinoma and endocervical adenocarcinoma (CESC).

Although the study is interesting, the authors have used many terminologies that are confusing, confounding, erroneous and out or context. Moreover, there appears to be flaw in the interpretation and techniques used.

To cite a few :

1. What is CESC? Is this a standard terminology?

2. Line 104: para-cancer and page 112: para-carcinoma

3. How was the histopathological diagnosis of CESC made. Was any histopathologist include in the study?

4. ESTIMATE scores were based on samples from 11 different tumour types which does not include cervical carcinoma; has any other study used it?

5. Line 149-152: Followingly, tissues were cultured with primary antibody anti-MLK4 (GXP326155, 1:200,

GENXSPAN，USA) at 4°C overnight. After washed with tris buffered saline (TBS) solution containing 0.025% Triton X-100 twice, the samples were co-cultured with second antibody and stained with DAB (Biosharp, BL732A) at room temperature for about 10 min.

How is it possible to co-culture a dead tissue (paraffin embedded tissue)?

These are some of the examples.

I would recommend to completely rewrite the manuscript.

6. PLOS authors have the option to publish the peer review history of their article (what does this mean?). If published, this will include your full peer review and any attached files.

Reviewer #1: No

Reviewer #2: No

---

## [Author Response · Author response to Decision Letter 0]

3 Jul 2023

Response to Reviewers

Title: MLK4 as an immune marker and its correlation with immune infiltration in Cervical squamous cell carcinoma and endocervical adenocarcinoma (CESC)

Manuscript ID: PONE-D-23-02890

Dear Editor,

On behalf of my co-authors, we thank you very much for giving us the opportunity to revise the manuscript. We also deeply appreciate the reviewers' time and effort to help us to improve the manuscript. Firstly,we have addressed some minor occurrence of overlapping text with the following previous publication（-https://doi.org/10.18632/aging.102514, - https://doi.org/ 10.1111/ejh.13895）.We also take into consideration that the majority of the research is an analysis of public databases and clinical research, and that additional molecular experiments are required to confirm its biological role in cervical cancer. As a result, We send our members to a molecular biology lab with cell experimental circumstances to learn and complete several functional validation investigations in recent months. A new manuscript has been added that includes related results. All experiments were recently finished, and all original data were available. 

We have considered the comments and suggestions seriously and tried our best to revise the manuscript. The revised portions are highlighted in yellow. We hope that our edits and responses satisfactorily address all the issues and concerns which you and the reviewers have noted. Besides,We are very sorry that this manuscript has been stored on a preprint server(https://doi.org/10.21203/rs.3.rs-2862240/v1) , the latest original manuscript was not accepted by any Journal.Also,we have required the journal to remove the preprint manuscript and are waiting for a reply. We hope this will not affect our submission.Thank you for receiving our manuscript and considering it for publication. We appreciate your time and look forward to your response.

Sincerely,

Li Hong, 

Gynecology Department , 

Renmin Hospital of Wuhan University, 

Jiefang Road 238, Wuhan, Hubei 430060, P.R. of China

E-mail: drwhdxrmyyhongli@163.com

Tel/ Fax: +86 27 88041911

Please find the following Response to the comments of referees:

Reviewers' Comments:

1).I recommend professional English editing.

Response: Thank you for the suggestion. With the assistance of Editage (www.editage.cn) and experts for English language editing, we improved the language in this manuscript.We believe this improvement considerably raised the paper's quality.

2)Abstract: Please include numbers such as sample sizes of TCGA and array data, p values etc.

Response: Thank you for the valuable comments. We have performed bioinformatics analysis of RNA sequencing (RNA-Seq) data from individuals with Cervical squamous cell carcinoma and endocervical adenocarcinoma(CESC) from the TCGA (https://cancergenome.nih.gov/.), which contais 306 CESC tissues and 3 peritumoral tissue samples up to September 18, 2021. The Deseq2 package was used to identify differentially expressed genes (DEGs) between tumor and normal samples (Benjamini-Hochberg corrected p-value 0.05 and log2 fold change |2|). (This explanation has been added to lines 328–332 of the Abstract.) (This description has been added to the Abstract, lines 52-54 and Method, lines 128-133.)

3)Material and Method:Can you describe ESTIMATE briefly?

Response: Thank you for the suggestion.ESTIMATE (Estimation of STromal and Immune cells in MAlignant Tumor tissues using Expression data) is a computational tool that estimates the abundance of stromal and immune cells in tumor tissues based on gene expression data. It uses a gene signature-based approach to infer the proportion of stromal and immune cells in the tumor microenvironment, which can provide insights into the role of these cells in tumor progression and response to therapy. In cervical cancer research, ESTIMATE has been used to analyze gene expression data from cervical cancer tissues and to investigate the association between the tumor microenvironment and patient outcomes.

 ESTIMATE algorithm is based on single sample Gene Set Enrichment Analysis and generates three scores:

1) stromal score (that captures the presence of stroma in tumor tissue),

2) immune score (that represents the infiltration of immune cells in tumor tissue)

3) estimate score (that infers tumor purity).

So far, the ESTIMATE R package was migrated to R-Forge, available on the website (https://r-forge.r-project.org/R/?group_id=2237).

4）How was IHC scored? What is a low or high IHC expression (table 2)? Please define them in the text.

Response: Thank you for the valuable comments. In fact, wwe have used the software QuantCenter 2.2 to analyze the immunohistochemistry result of MLK4. The application identifies positive dyes based on an automatic color separation method, through which individual positive expression pixels can be counted and classified according to their intensity and threshold range. The ratio of positive to negative pixels can then be used to determine the H-Score. Positive area ratio=Positive area/total tissue area.(This description has been added to the Method, lines 179-183.)

5）I cannot download supplementary data. Can you submit an IHC picture?

Response: Yes, sure. The Picture of cervical cancer pathology microarray with MLK4 was shown below.

6) Can you submit Kaplan-Meier curves?

Response: Thank you for the valuable comments. The Kaplan-Meier curves for Disease-Free Survival and Overall Survival have been included (new Figures 3A and B).Disease-free and overall survival rates are shown on Kaplan-Meier curves ?for?CESC patients with low and high MLK4 levels, respectively,recorded in TCGA array data. As result has shown, low MLK4 expression was correlated with good DFS. (Figure 3A) (Logrank p=0.014, p(HR) = 0.016) .In TCGA data, low MLK4 expression was also correlated with good OS. (Figure 3B) (Logrank p=0.45, p(HR) = 0.45) .(This description has been added to new figure 3B and the Result, lines 268-270.)

7)Can you make a table/diagram to summarize the interaction of MLK4 with immunological molecules?

Response: Thank you for the valuable comments. We have collected reports relating the MLK4 gene to a specific immune-mediated molecule. The table below summarizes the relationship between MLK4 and the immune molecules TLR4, TNF, IFN, IL-6, IFN-γ, and TGF-β.(This description has been added to Table 3 and the Result, lines 355-358.)

Immunological Molecules Interaction with MLK4 Effect PMID

Toll-like receptor 4 (TLR4) Directly activates MLK4 Induces pro-inflammatory cytokine production, activates NF-κB, and promotes dendritic cell maturation and antigen presentation 21602844

Tumor necrosis factor alpha (TNF-α) Activates MLK4 via TNFR1 Induces apoptosis and inflammation, promotes macrophage activation and cytokine production, and enhances T cell activation and proliferation 21602844

Interleukin-1 type 1 receptor(IL-1R1) Activates MLK4 via IL-1R1 Induces pro-inflammatory cytokine production, activates NF-κB and MAPK pathways, and promotes neutrophil recruitment and activation 33879157

Interferon gamma (IFN-γ) Activates MLK4 Induces pro-inflammatory cytokine production, enhances antigen presentation and T cell activation, and promotes macrophage polarization towards M1 phenotype 15294995

Interleukin-6 (IL-6) Indirectly activates MLK4 via STAT3 Induces pro-inflammatory cytokine production, promotes B cell differentiation and antibody production, and enhances Th17 cell differentiation and autoimmunity 15294995

 Interleukin-17receptor B

(IL-17RB) MLK4 specifically phosphorylates IL-17RB at Y447 Induces pro-inflammatory cytokine production, IL-17RB forms a homodimer and recruits MLK4, a dual kinase, to phosphorylate it at tyrosine-447 upon treatment with IL-17B 33658352

*abbreviations:MLK4 stands for Mixed Lineage Kinase 4. NF-κB stands for Nuclear Factor kappa B. MAPK stands for Mitogen-Activated Protein Kinase. Th17 stands for T helper 17.TGF-βstands forTransforming growth factor beta. 

8)What is para tumor tissue?

Response: Thank you for the valuable comments.We intended to describe the tissue nearby cancer tissue with the word "para-tumor tissue".However, this translation is not accurate with the help of experts, we replace "para-tumor tissue" with "peritumoral" in this manuscript.

9)Can you give the exact number of SCC and adenocarcinoma samples?

Response: Thank you for the valuable comments. There were 255 Squamous cell carcinoma samples and 51 endocervical adenocarcinoma samples in TCGA database. TMA data sets contain 75 Squamous cell carcinoma tiusse samples and 5 endocervical adenocarcinoma tissue samples.

10)How many normal samples were used in TCGA and TMA data sets?

Response: Thank you for the valuable comments. There were 306 CESC tissues and 3 peritumoral tissue samples in TCGA database. TMA data sets contain 80 CESC tissues and 80 peritumoral tissue samples.

11)What is small RNA, do you mean miRNA?

Response: Thank you for the valuable comments. In this manuscript, small RNA stands for a class of RNA molecules without protein-coding ability, such as miRNA, piRNA, tsRNA (tRF&tiRNA), sRNA and snoRNA, etc.

Referee#2

The authors have studied the utility of MLK4 as an immune marker and correlated the same with immune cell infiltration in cervical squamous cell carcinoma and endocervical adenocarcinoma (CESC).

Although the study is interesting, the authors have used many terminologies that are confusing, confounding, erroneous and out or context. Some concerns should be addressed:

1)What is CESC? Is this a standard terminology?

Response: Thank you for the suggestions. Cervical cancers are named after the type of cell where the cancer started. The two main types are Squamous cell carcinoma and Adenocarcinoma. Most cervical cancers (up to 90%) are squamous cell carcinomas. These cancers develop from cells in the ectocervix. However, Cervical adenocarcinomas develop in the glandular cells of the endocervix. Besides, squamous cell carcinoma, which develops in the thin, flat, squamous cells that line the vagina, and adenocarcinoma, which arises in the glandular cells in the vagina that secrete mucus.Squamous cell carcinomas (SCC) and adenocarcinomas (AC) are the two primary histological variants, and they both resemble the cell morphology of the exocervix and endocervix, respectively(This description has been added to the Introduction, lines 77-79.)

 We also found certain references based on CESC TCGA array data, which are relative to tumor microenvironment characteristic of cervical cancer.

【1】Chen, Qian et al. "Identification of a tumor microenvironment-related gene signature to improve the prediction of cervical cancer prognosis." Cancer cell international vol. 21,1 182. 25 Mar. 2021, doi:10.1186/s12935-021-01867-2

【2】Deng, Yali et al. "Tumor purity as a prognosis and immunotherapy relevant feature in cervical cancer." Aging vol. 13,22 (2021): 24768-24785. doi:10.18632/aging.203714

【3】Shi, Yujing et al. "Identification of Immune and Hypoxia Risk Classifier to Estimate Immune Microenvironment and Prognosis in Cervical Cancer." Journal of oncology vol. 2022 6906380. 17 Oct. 2022, doi:10.1155/2022/6906380

2)Line 104: para-cancer and page 112: para-carcinoma

Response: Thank you for the valuable comments. We intended to describe the tissue nearby cancer tissue with the word "para-tumor tissue".However, this translation is not accurate. With the help of experts, we replace "para-tumor tissue" with "peritumoral" in this manuscript.

3)How was the histopathological diagnosis of CESC made. Was any histopathologist include in the study?

Response: Thank you for the suggestion. Commercialized tissue microarray provides relative pathological information, they have compliant sample collection standards, and the pathologist is involved in the tissue diagnosis. We also invited the histopathologist of our hospital to participate in the diagnosis of tissue microarray.

4)ESTIMATE scores were based on samples from 11 different tumour types which does not include this cancer; has any other study used it?

Response: Thank you, this is a good question. The levels of infiltrating stromal and immune cells were predicted by computing stromal and immune scores by performing a single-sample gene set enrichment analysis (ssGSEA), which forms the basis for inferring the ESTIMATE Score of tumor purity in tumor tissue. The algorithm model can theoretically be used to analyze different cancer types. Additionally, we noticed that the ESTIMATE database (https://bioinformatics.mdanderson.org/estimate/) containstable of Summary of cancer type expression data that is available for cervical cancer.

Besides, We discovered several references between the ESTIMATE algorithm and cervical cancer. Lirong Yang et al(DOI: 10.1016/j.gene.2020.145119) used ESTIMATE algorithm and the Cell type Identification By Estimating Relative Subsets Of known RNA Transcripts (CIBERSORT）deconvolution algorithm to quantify the fraction and infiltration of 22 types of immune cells in cervical cancer. Additionally, they discovered that CCR5 and the CXCL9, -10, -11/CXCR3 axis may provide a novel target for the therapy of cervical cancer (DOI: 10.1016/j.gene.2020.145119).

Deng et al ( DOI： 10.18632/aging.203714 ) also used ESTIMATE algorithm to calculate the tumor purity of cervical cancer. Therefore, ESTIMATE has been utilized in cervical cancer research to analyze the relationship between the tumor microenvironment and patient outcomes and to assess gene expression data from cervical cancer tissues. (This description has been added to the Introduction, lines 105-109）.

We also consider that the majority of the study consists of an analysis of public databases and clinical research, and that it requires additional molecular investigations to confirm its biological role in cervical cancer. In recent months, we have added cell molecular experiments to determine the biological behavior of MLK4 in cervical cancer cells. In addition, vitro assays were carried out to explore the biological behavior of MLK4 in C33A cells. Bioinformatics analysis have revealed that the high expression of MLK4 was associated with immune checkpoint CD274, CTLA4, LAG3 and IDO1 were positively correlated with immune-promoting genes, such as CD86 and CD80.qPCR further confirmed this result. The expression of the inflammatory cytokinesIL-1β(P<0.05), TNF-α(P<0.01)and IL-6 (P<0.05) was significantly increased by the silencing of MLK4. The results of cell assays indicate that knocking down MLK4 would inhibit the expression of conventional biochemical markers CEA，AFP and HCG. MLk4 may therefore play a crucial function in cervical cancer.MLk4 may play a critical role in Cervical cancer.(This description has been added to the Results, lines 316-376.)

5)Line 149-152: Followingly, tissues were cultured with primary antibody anti-MLK4 (GXP326155, 1:200, GENXSPAN，USA) at 4°C overnight. After washed with tris buffered saline (TBS) solution containing 0.025% Triton X-100 twice, the samples were co-cultured with second antibody and stained with DAB (Biosharp, BL732A) at room temperature for about 10 min.

How is it possible to co-culture a dead tissue (paraffin embedded tissue)?

Response: Thank you for the suggestion. We intended to describe the process of IHC in the step of incubating with antibody. However, this translation is not accurate with the help of experts, we replace "co-culture" with "incubated" in this manuscript.(This description has been added to the Results, lines 173-184.)We found that there was some translation problems in this manuscript. With the assistance of Editage (www.editage.cn) and experts for English language editing, we improved the language in this manuscript.We believe this improvement considerably raised the paper's quality.

---

## [Decision Letter · Decision Letter 1]

9 Aug 2023

MLK4 as an immune marker and its correlation with immune infiltration in Cervical squamous cell carcinoma and endocervical adenocarcinoma(CESC)

PONE-D-23-02890R1

Dear Dr. Li,

We’re pleased to inform you that your manuscript has been judged scientifically suitable for publication and will be formally accepted for publication once it meets all outstanding technical requirements.

Kind regards,

Parikshaa Gupta

Academic Editor

PLOS ONE

Additional Editor Comments (optional):

The authors have satisfactorily modified the manuscript as per the suggestions and it is now acceptable for publication.

Reviewers' comments:

Reviewer's Responses to Questions

**Comments to the Author**

1. If the authors have adequately addressed your comments raised in a previous round of review and you feel that this manuscript is now acceptable for publication, you may indicate that here to bypass the “Comments to the Author” section, enter your conflict of interest statement in the “Confidential to Editor” section, and submit your "Accept" recommendation.

Reviewer #2: All comments have been addressed

2. Is the manuscript technically sound, and do the data support the conclusions?

Reviewer #2: Yes

3. Has the statistical analysis been performed appropriately and rigorously? 

Reviewer #2: I Don't Know

4. Have the authors made all data underlying the findings in their manuscript fully available?

Reviewer #2: Yes

5. Is the manuscript presented in an intelligible fashion and written in standard English?

Reviewer #2: Yes

6. Review Comments to the Author

Reviewer #2: I appreciate the authors for addressing the concerns raised (including the English language editing).

I do not have further queries.

7. PLOS authors have the option to publish the peer review history of their article (what does this mean?). If published, this will include your full peer review and any attached files.

Reviewer #2: No

---

## [Editor Report · Acceptance letter]

11 Aug 2023

PONE-D-23-02890R1 

MLK4 as an immune marker and its correlation with immune infiltration in Cervical squamous cell carcinoma and endocervical adenocarcinoma(CESC) 

Dear Dr. Hong:

I'm pleased to inform you that your manuscript has been deemed suitable for publication in PLOS ONE. Congratulations! Your manuscript is now with our production department. 

Kind regards, 

on behalf of

Dr. Parikshaa Gupta 

Academic Editor

PLOS ONE